# Peroxisomal Alterations in Prostate Cancer: Metabolic Shifts and Clinical Relevance

**DOI:** 10.3390/cancers17132243

**Published:** 2025-07-04

**Authors:** Mohamed A. F. Hussein, Celien Lismont, Hongli Li, Ruizhi Chai, Frank Claessens, Marc Fransen

**Affiliations:** 1Laboratory of Peroxisome Biology and Intracellular Communication, Department of Cellular and Molecular Medicine, KU Leuven, 3000 Leuven, Belgium; mohamed.hussein@kuleuven.be (M.A.F.H.); celien.lismont@kuleuven.be (C.L.); hongli.li@kuleuven.be (H.L.); ruizhi.chai@kuleuven.be (R.C.); 2Department of Biochemistry, Faculty of Pharmacy, Assiut University, Asyut 71515, Egypt; 3Laboratory of Molecular Endocrinology, Department of Cellular and Molecular Medicine, KU Leuven, 3000 Leuven, Belgium; frank.claessens@kuleuven.be

**Keywords:** peroxisomes, prostate cancer, androgen receptor, fatty acid oxidation, ether lipids, metabolic rewiring, redox homeostasis

## Abstract

Peroxisomes have attracted growing interest for their potential role in prostate cancer, a disease characterized by metabolic features distinct from the glycolysis-dominated metabolism typical of most other tumors. These organelles play a crucial role in lipid and redox metabolism, and alterations in their function have been associated with androgen receptor signaling, prostate cancer progression, and therapy resistance. Despite their importance, the role of peroxisomes in prostate cancer biology remains underexplored. This review examines peroxisomal metabolic reprogramming in prostate cancer, integrates current research on the topic, identifies critical knowledge gaps, and emphasizes the importance of clarifying the organelle’s causal role in tumorigenesis. This includes distinguishing peroxisome-specific contributions and developing accurate experimental models. Advancing our understanding in these areas could lead to the discovery of novel biomarkers and therapeutic strategies, ultimately improving outcomes for prostate cancer patients.

## 1. Introduction

Cancer is characterized by uncontrolled cell proliferation, enhanced migration and invasion, resistance to cell death, and immune evasion—hallmarks that, along with metabolic rewiring, collectively drive tumor progression [1]. Organellar dysfunction plays a critical role in these processes, as organelles regulate key metabolic pathways, including fatty acid (FA) metabolism and oxidative phosphorylation, as well as vital cellular processes such as cell fate determination, redox signaling, and calcium homeostasis [2,3,4,5,6]. For example, in many human tumors, dysregulations in mitochondrial respiration play a critical role in cancer progression and metastasis [7]. In addition, functional deficits in outer membrane permeabilization or mitochondrial permeability transition enable malignant cells to evade regulated cell death [8]. Likewise, endoplasmic reticulum (ER) stress and the activation of the unfolded protein response have been linked to tumor development [9]. In addition, alterations in peroxisome biogenesis and function have been reported across various cancer types [10].

Peroxisomes are single-membrane-bound organelles that play a crucial role in various anabolic and catabolic processes [11]. In mammals, they are indispensable for the oxidative breakdown of very long-chain fatty acids (VLCFAs) and branched-chain fatty acids (BCFAs). They also contribute to the biosynthesis of key lipid species, including the formation of the ether bond in all ether lipids (e.g., plasmalogens) and the production of docosahexaenoic acid (DHA; C22:6 n-3). In the liver, peroxisomes participate in the detoxification of glyoxylate by converting it into glycine, the synthesis of primary bile acids (chenodeoxycholic acid and cholic acid), and their conjugation with glycine or taurine for excretion into bile [11]. Beyond their metabolic roles, peroxisomes are important for redox homeostasis [12] and the regulation of signaling molecules and pathways [11]. They harbor various oxidases (e.g., acyl-CoA oxidases, D-amino acid oxidases, L-α-hydroxy acid oxidases, and polyamine oxidases) that generate hydrogen peroxide (H_2_O_2_) as a metabolic by-product. Under conditions of sufficient substrate availability, peroxisomes can consume up to 20% of cellular oxygen and account for about 35% of the cytosolic H_2_O_2_ [13,14]. Oxidative stress is mitigated by peroxisomal catalase (CAT), a key antioxidant enzyme responsible for degrading H_2_O_2_ [12].

Prostate cancer (PCa) is the second most diagnosed cancer worldwide and the fifth leading cause of cancer-related mortality in men [15]. The androgen receptor (AR), a ligand-dependent transcription factor, serves as the principal driver of PCa initiation and progression [16]. Consequently, androgen deprivation therapy is the cornerstone treatment for metastatic PCa. However, resistance to hormonal therapy inevitably arises in most patients, resulting in the development of castration-resistant PCa (CRPC) [17]. Alterations in AR signaling are closely linked to metabolic reprogramming that supports tumor growth and disease progression [18]. While the Warburg effect, characterized by increased glycolytic activity, was the first metabolic hallmark identified in cancer, primary PCa differs in that it exhibits relatively low glycolytic rates. Instead, it depends predominantly on enhanced lipogenesis and oxidative phosphorylation [19,20,21,22]. This distinct metabolic profile helps explain the limited effectiveness of fluorodeoxyglucose imaging in detecting localized PCa [23]. In contrast, neuroendocrine PCa, an aggressive AR-negative subtype, adopts a distinct well-established PCa glycolytic phenotype [24]. These divergent metabolic characteristics underscore the unique nature of PCa among solid tumors [25].

Emerging evidence suggests that the dysregulation of peroxisomal function contributes to PCa initiation, progression, and therapeutic response by disrupting lipid metabolism and redox balance—two key hallmarks of PCa biology. This comprehensive literature review is guided by the working hypothesis that alterations in peroxisome function are not merely correlative but play a mechanistic role in PCa pathogenesis and treatment outcomes. We critically examine how peroxisomal dysfunction influences tumor biology with particular emphasis on its interaction with AR signaling.

## 2. Methods

### 2.1. Data Sources and Search Strategy

A literature search was conducted using the PubMed database with the following search string: peroxisomes AND (prostate cancer OR prostate adenocarcinoma OR androgen receptor). In addition, reference lists of relevant articles were manually screened for potentially eligible studies. No date restrictions were applied, and literature published up to June 2025 was included.

The mRNA expression profiles of primary PCa and normal prostate tissue were retrieved from The Cancer Genome Atlas Program (TCGA) via the University of Alabama at Birmingham Cancer Data Analysis Portal (UALCAN; https://ualcan.path.uab.edu/; accessed on 12 March 2025) [26].

### 2.2. Eligibility Criteria

Included sources consisted of peer-reviewed articles, book chapters, reviews, in vitro and in vivo studies, clinical studies, and systematic analyses, provided they were published in English. Only studies investigating intracellular changes resulting from peroxisomal alterations in PCa were considered.

### 2.3. Exclusion Criteria

Letters, proceedings, conference abstracts, and editorials were excluded, as were studies focusing on changes resulting from peroxisomal alterations in patient blood samples.

### 2.4. Study Screening and Selection Process

Two authors (MH and MF) independently screened articles for eligibility according to the inclusion criteria. After the initial title and abstract screening, both investigators reviewed the full texts of the selected studies and reached inclusion decisions by mutual consensus.

### 2.5. Data Extraction and Processing

Data extraction was guided by the list of peroxisome-related proteins reported in Yifrach et al. (2018) [27], supplemented by recent literature identifying proteins associated with peroxisomes or peroxisomal maintenance. For the UALCAN data, mRNA expression values (transcripts per million) and corresponding *p*-values were used to calculate log_2_ fold changes relative to controls, along with the associated log_10_ *p*-values. It is important to note that the UALCAN database does not provide explicit fold-change (FC) cutoff values, as its primary focus is to facilitate an interactive exploration of gene expression data rather than to perform standardized differential expression analysis using strict statistical thresholds. The volcano plots were generated using GraphPad Prism Software (version 10.2.2; San Diego, CA, USA). 

## 3. Results

### 3.1. Peroxisome Abundance and Prostate Cancer

Peroxisome levels are tightly regulated through a balance between their formation and degradation. Peroxisomes can be generated via two primary pathways: de novo biogenesis and the growth and division of existing peroxisomes [11,28,29,30,31]. These processes are governed by peroxisome biogenesis factors known as peroxins (PEX). Peroxisomal degradation occurs through a selective autophagy process called pexophagy, which can proceed through either ubiquitin-dependent or ubiquitin-independent mechanisms [32,33,34]. Pexophagy can be triggered by various cellular stressors, including oxidative stress, nutrient depletion, hypoxia, and defects in peroxisome biogenesis [32].

In localized PCa, the mRNA expression levels of several proteins involved in peroxisome biogenesis, proliferation, and degradation differ markedly between malignant and adjacent non-malignant tissues (Figure 1A). Similar patterns are observed at the protein level in PCa cell lines (22Rv1, LNCaP, and PC3) when compared to the non-malignant RWPE-1 cell line [35]. Notably, proteins within the same pathway (e.g., PEX10 and PEX14) often exhibit divergent expression patterns, while proteins from functionally opposing pathways (e.g., PEX19 and OPTN) may show similar expression trends. This complexity underscores the limitations of current data in clearly elucidating the relationship between peroxisome abundance and PCa malignancy.

An emerging regulatory factor in this context is AR signaling. While the role of AR in peroxisome homeostasis in PCa remains not fully understood, recent studies have begun to shed light on potential mechanisms. Feng et al. (2024) identified the *PEX10* gene as a direct transcriptional target of AR [36]. In their study, AR knockdown in C4-2 and LNCaP cells resulted in elevated levels of reactive oxygen species (ROS), indicating increased oxidative stress. Notably, the ectopic overexpression of PEX10 following AR depletion restored peroxisome abundance, reduced ROS levels, and promoted cell proliferation. Moreover, silencing PEX10 increased the sensitivity of PCa cells to ferroptosis inducers both in vitro and in vivo, suggesting a promising therapeutic angle. These findings highlight the critical role of the AR-mediated regulation of PEX10 in maintaining peroxisome dynamics and oxidative stress responses, which could have implications for PCa progression and treatment. Supporting this, the AR signaling inhibitor enzalutamide was shown to suppress PEX10 expression, revealing an additional mechanism by which it may exert its antitumor effects [36]. Consistent with these findings, gene set enrichment analysis revealed that the pathway peroxisome was downregulated in localized high-risk patients treated with the poly (adenosine diphosphate ribose) polymerase inhibitor fuzuloparib in combination with the AR pathway inhibitor abiraterone [37].

Several studies have investigated the role of the monocarboxylate transporter 2 (MCT2) in regulating peroxisome morphology and function [38,39,40]. Valença et al. (2015) found that MCT2 is more prominently localized to peroxisomes in PCa cells (22Rv1 and PC3) compared to the non-malignant prostate cell line PNT1A [38]. In 22Rv1 cells, MCT2 overexpression resulted in an increase in peroxisomal surface area but a reduction in peroxisome number [39]. Interestingly, MCT2 knockdown did not result in a corresponding increase in peroxisome numbers [39], suggesting that MCT2’s role in peroxisome regulation is more complex than initially assumed. In addition, while peroxisome morphology was similar between PC3 and PNT1A cells, 22Rv1 cells exhibited more elongated and clustered peroxisomes, indicating that peroxisome shape may depend on the cellular context [38]. Notably, the expression of MCT2, encoded by the *SLC16A7* gene, is regulated by AR, and the selective demethylation of an internal SLC16A7 promoter appears to be a recurrent event in independent PCa cohorts [40]. Finally, MCT2 knockdown was found to reduce the proliferation of 22Rv1 and PC3 cells [40].

These findings underscore the complex regulation of peroxisome abundance in PCa, suggesting that factors such as PEX10 and MCT2 may play a role in modulating peroxisome dynamics and could contribute to the development of new therapeutic strategies.

### 3.2. Peroxisomes, Ether Lipids, and Prostate Cancer

#### 3.2.1. Structure and Function of Ether Lipids

Ether lipids are uniquely characterized by the presence of a long-chain fatty alcohol linked to the *sn*-1 position of the glycerol backbone via an ether bond. This ether linkage can be either saturated—referred to as plasmanyl or alkyl ether lipids—or unsaturated—known as plasmenyl or alkenyl ether lipids—with the latter typically featuring a cis-vinyl configuration (Figure 2). The *sn*-2 position may be unmodified or esterified with a FA, while the *sn*-3 position can be free, esterified with an FA, or linked to a phosphate group. When present, the phosphate group can further bind to a nitrogenous base, most commonly ethanolamine or choline. This structural versatility contributes to the generation of a wide variety of ether lipid species [41].

Plasmalogens, the most abundant class of ether lipids, belong to the alkenyl ether lipid subclass. Their typical structure includes (i) a fatty alcohol (C16:0, C18:0, or C18:1) at the *sn*-1 position, (ii) a long-chain polyunsaturated fatty acid (PUFA; e.g., arachidonic acid (C20:4) or DHA (C22:6)) or a monounsaturated fatty acid (MUFA; e.g., oleic acid (C18:1)) at the *sn*-2 position, and (iii) a phosphodiester bond at the *sn*-3 position, which links the glycerol backbone to the polar head group, typically ethanolamine or choline [41]. Phospholipase 2 can hydrolyze the *sn*-2 ester bond, releasing bioactive lipids such as arachidonic acid and DHA [42]. Plasmalogens account for approximately 20% of total phospholipids in humans, with the highest concentrations found in the brain, heart, spleen, and neutrophils [43].

Ether lipids play several vital biological roles. They are key components of cellular membranes, particularly in myelin sheaths and neutrophils [44]. In addition to their structural functions, ether lipids influence membrane dynamics [42], participate in various cellular signaling pathways [45], and contribute to the cell’s antioxidant defenses [46].

#### 3.2.2. The Role of Peroxisomes in Ether Lipid Synthesis

Ether lipid synthesis begins with the esterification of the hydroxyl group of dihydroxyacetone-3-phosphate (DHAP; also known as glycerone-3-phosphate) with a fatty acyl-CoA, resulting in the formation of 1-acyl-dihydroxyacetone-3-phosphate (1-acyl-DHAP) and the release of coenzyme A (CoASH) as a by-product (Figure 2). This critical first step is catalyzed by glycerone-3-phosphate *O*-acyltransferase (GNPAT), also referred to as dihydroxyacetone-3-phosphate *O*-acyltransferase (DHAPAT). DHAP, a glycolytic intermediate, can either be imported into peroxisomes from the cytosol or synthesized within peroxisomes from glycerol-3-phosphate by the action of peroxisomal glycerol-3-phosphate dehydrogenase 1 [47,48].

Next, the acyl moiety of 1-acyl-DHAP is replaced by a fatty alcohol, resulting in the formation of 1-*O*-alkyl-DHAP through an ether bond. This reaction is catalyzed by 1-alkyl-glycerone-3-phosphate synthase (AGPS), also known as 1-*O*-alkyl-dihydroxyacetone-3-phosphate synthase (ADHAPS). Both GNPAT and AGPS are peroxisomal enzymes, highlighting the essential role of peroxisomal activity in the formation of ether bonds in all ether lipids [49]. The fatty alcohol used in the AGPS-catalyzed reaction is produced by fatty acyl-CoA reductase (FAR), which exists in two isoforms (FAR1 and FAR2) that are anchored by their C-terminus in the peroxisomal membrane with their catalytic domains oriented toward the cytosol [50].

The product of the AGPS reaction, 1-*O*-alkyl-DHAP, is subsequently transported to the ER, where it is reduced to 1-*O*-alkyl-glycerol-3-phosphate (1-*O*-alkyl-G3P) by acyl/alkyl-dihydroxyacetone-3-phosphate reductase (ADHAPR) [51]. Although a peroxisomal isoform of this enzyme, dehydrogenase/reductase 7B (DHRS7B), also known as PexRAP (peroxisomal reductase activating PPARγ), has been identified, it is incapable of catalyzing this specific reduction [51,52]. All subsequent steps in ether lipid synthesis also take place in the ER.

Lipidomic analyses of mouse livers lacking the peroxisomal membrane protein (PMP) PXMP4 revealed reduced levels of alkyldiacylglycerols, a class of neutral ether lipids, with the most pronounced decrease observed in species containing PUFAs [53]. These findings suggest that PXMP4 may have an as-yet-undefined role in ether lipid metabolism.

#### 3.2.3. Peroxisomal Ether Lipid Metabolism in Prostate Cancer

Ether lipid metabolism has become an area of growing interest in PCa research, with particular focus on AGPS, a key enzyme in the biosynthesis of ether lipids. Notably, AGPS expression is significantly downregulated in PCa tissues compared to adjacent non-malignant tissues, at both the mRNA and protein levels, as demonstrated by clinical sample analyses [54] and corroborated by TCGA data (Figure 1B). These findings suggest a potential tumor suppressive role for AGPS in vivo and highlight the need for further investigation into its functional relevance in PCa.

Similarly, PXMP4—a PMP recently implicated in gastric cancer cell proliferation, invasion, and migration via the phosphatidylinositol 3-kinase (PI3K)/Akt signaling pathway [55]—also exhibits significantly reduced mRNA expression in PCa tissues (Figure 1B). This observation is consistent with reports of PXMP4 DNA hypermethylation during PCa progression and in LNCaP-derived models [56,57]. In contrast, the expression levels of other ether lipid biosynthetic enzymes (e.g., GNPAT, FAR1, and FAR2) remain unchanged in clinical specimens.

Despite the consistency of clinical findings, in vitro data reveal a more complex picture, particularly concerning AGPS. For example, AGPS expression has been reported to be elevated in the highly aggressive PC3 cell line, which also displays higher ether lipid levels compared to the less aggressive LNCaP cells [58]. Furthermore, the supplementation of PC3 cells with hexadecylglycerol, a precursor of ether lipids, further increases ether lipid levels both intracellularly and in exosomes [59]. In addition, PC3 cells divert fatty acid metabolism away from oxidative pathways, favoring instead the synthesis of ether-linked structural and signaling lipids, including platelet-activating factor [60].

Functionally, the pharmacological inhibition of AGPS in PC3 cells suppresses epithelial-to-mesenchymal transition (EMT), as indicated by an altered expression of E-cadherin, Snail, and matrix metalloproteinase-2 (MMP2). At higher inhibitor concentrations, AGPS suppression also impairs cell proliferation, suggesting a potential pro-tumorigenic role for AGPS in this specific cellular context [61]. Furthermore, extracellular vesicles derived from PC3 cells are enriched in ether-linked phosphatidylcholine, whereas those from RWPE-1 (non-malignant) and NB26 (malignant) cells contain higher levels of ether-linked phosphatidylethanolamine, reflecting distinct, cell type-specific lipid profiles [62].

Interestingly, AGPS expression has also been reported to be lower in multiple PCa cell lines (e.g., 22Rv1, DU145, C4-2, and PC3) compared to the non-malignant RWPE-1 cell line [54]. Proteomics analyses further confirm reduced AGPS protein levels in 22Rv1 and LNCaP cells, though not in PC3 cells [35]. The lipidomic profiling of PCa tissue revealed a decrease in two plasmalogen phosphatidylethanolamine species [63], along with a broader downregulation of both saturated and unsaturated ether-linked lipids [64]. Spatial lipidomics studies have also shown that host adipocytes possess higher levels of triacylglycerol ethers, plasmanyl cholines, and plasmenyl cholines compared to LNCaP xenograft tumors [65]. However, one study reported increased ether lipid levels in PCa tissues [66], suggesting potential heterogeneity among patient cohorts and experimental models.

Recent work by Zhang et al. has provided functional insights into the role of AGPS in PCa, showing that the overexpression of AGPS in 22Rv1 cells suppressed cell proliferation and colony formation while increasing levels of malondialdehyde, a marker of lipid peroxidation [54]. Conversely, AGPS knockdown in PC3 and DU145 cells enhanced tumorigenic behaviors and decreased malondialdehyde levels. These in vitro findings were corroborated in DU145 xenografts, where AGPS knockdown significantly accelerated tumor growth. Furthermore, AGPS overexpression in 22Rv1 cells sensitized them to ferroptosis, as evidenced by a decreased IC50 for the ferroptosis inducers ML210 and RSL3. In contrast, AGPS knockdown in PC3 and DU145 cells led to increased IC50 values, indicating greater resistance to ferroptosis.

Mechanistically, the E3 ligase MDM2 was found to bind and ubiquitinate AGPS, targeting the protein for proteasomal degradation [54]. MDM2 knockdown resulted in a dose-dependent increase in AGPS expression. In addition, TrkA, a tyrosine kinase receptor implicated in PCa, was shown to phosphorylate AGPS at tyrosine 451, which enhanced AGPS’s interaction with MDM2 and facilitated its degradation. The pharmacological inhibition of TrkA using Larotrectinib led to AGPS accumulation and sensitized PCa cells to ferroptosis both in vitro and in vivo [54].

Emerging evidence links ether lipids—particularly the peroxidation-prone plasmalogens—to ferroptosis sensitivity, despite their traditional roles as antioxidants [67,68]. Peroxisomes, which contribute to the synthesis of polyunsaturated ether phospholipids, have thus been identified as key modulators of ferroptosis, as demonstrated in ovarian and renal cancer cells [67]. Cancer cells may evade ferroptosis by downregulating enzymes such as AGPS and FAR1, leading to a significant reduction in vulnerable polyunsaturated ether phospholipid species. Similar findings have been observed in neurons and cardiomyocytes, where cellular differentiation correlates with increased ferroptosis resistance driven by ether lipid depletion [67]. Cui et al. (2021) further highlighted the critical role of peroxisome-derived ether lipids in regulating ferroptosis across multiple cancer types [68].

Despite growing evidence linking ether lipids and plasmalogens to ferroptosis sensitivity, several key questions remain. For example, how do plasmalogens, traditionally recognized for their antioxidant properties, contribute to ferroptosis induction? What are the distinct roles of alkyl- and alkenyl-ether lipids in cancer biology? How are oxidized ether lipids metabolized, and what is their precise function in ferroptotic signaling? A comprehensive understanding of the multifaceted roles of ether lipids in PCa cancer and other malignancies will require integrated approaches combining lipidomics, functional genomics, and in vivo studies. Notably, the discrepancies between in vivo and in vitro findings—particularly in PC3 cells—highlight the limitations of current experimental models and emphasize the need for context-dependent interpretation.

### 3.3. Peroxisomes, Fatty Acid Oxidation, and Prostate Cancer

Peroxisomes facilitate two distinct types of FA oxidation: β-oxidation and α-oxidation. Genetic defects that impair these pathways are associated with a variety of inherited metabolic disorders, including X-linked adrenoleukodystrophy, acyl-CoA oxidase 1 deficiency, D-bifunctional protein deficiency, and Refsum disease [69].

While both mitochondria and peroxisomes are involved in FA catabolism, their roles differ significantly. Mitochondrial β-oxidation primarily serves energy production, whereas peroxisomal FA oxidation contributes to FA shortening, detoxification, and the synthesis of bioactive lipids [70]. Notably, peroxisomes can also compensate for mitochondrial FA oxidation when the latter pathway is impaired or overwhelmed [71].

A key distinction between mitochondria and peroxisomes lies in their substrate import mechanisms. FAs enter mitochondria as acylcarnitines, whereas they are imported into peroxisomes as acyl-CoAs via three ATP-binding cassette subfamily D transporters: ABCD1, ABCD2, and ABCD3. These transporters exhibit overlapping but distinct substrate specificities. ABCD1 primarily transports saturated VLCFA-CoAs; ABCD3 (also known as PMP70) is responsible for importing CoA esters of BCFAs, dicarboxylic acids, and C27-bile acid intermediates [70]; and ABCD2 is believed to transport both saturated and unsaturated VLCFA-CoAs [72]. In addition, the PMP acyl-CoA binding domain containing 5 (ACBD5) facilitates lipid transfer from the ER to peroxisomes by tethering the two organelles through its interaction with vesicle-associated membrane protein-associated protein B (VAPB) [73].

#### 3.3.1. β-Oxidation

Peroxisomal β-oxidation initiates the breakdown of a broad range of FA substrates, including VLCFAs, BCFAs, and C27-bile acid intermediates such as dihydroxycholestanoic acid (DHCA) and trihydroxycholestanoic acid (THCA), as well as certain MUFAs, PUFAs, and dicarboxylic acids [11]. This pathway typically involves four key steps: (1) dehydrogenation (first oxidation); (2) the hydration of the double bond; (3) dehydrogenation (second oxidation), and (4) thiolytic cleavage, which produces acetyl-CoA (for non-BCFAs) or propionyl-CoA (for BCFAs), along with a shortened acyl-CoA chain (Figure 3).

The initial oxidation step is catalyzed by acyl-CoA oxidases (ACOX) 1-3, which convert acyl-CoAs into Δ^2^-trans-enoyl-CoAs. Each enzyme has specific substrate preferences: ACOX1 targets straight-chain VLCFA-CoAs; ACOX2 oxidizes BCFA-CoAs (e.g., pristanoyl-CoA, DHC-CoA, and THC-CoA) in the liver and kidney; and ACOX3 acts on BCFA-CoAs (excluding the bile acid intermediates DHC-CoA and THC-CoA) in other tissues [74]. For BCFAs, a critical enzyme is α-methylacyl-CoA racemase (AMACR), which converts the *R*-isomer of 2-methyl-branched CoA esters to the *S*-isomer, making them suitable for oxidation by ACOXs (Figure 4A). AMACR localizes to both peroxisomes and mitochondria [75,76,77].

Next, the hydration and dehydrogenation steps are catalyzed by L- and D-bifunctional proteins. The D-bifunctional protein (DBP), also known as multifunctional protein 2 (MFP2) and encoded by the *hydroxysteroid 17-beta dehydrogenase 4* (*HSD17B4*) gene, first hydrates the double bond of Δ^2^-trans-enoyl-CoAs to form 3-hydroxyacyl-CoAs and then oxidizes them to 3-ketoacyl-CoAs. HSD17B4 is specific for Δ^2^-trans-enoyl-CoAs derived from the CoA esters of VLCFAs, pristanic acid, DHCA, and THCA. In contrast, the L-bifunctional protein (LBP), also known as multifunctional protein 1 (MFP1) and encoded by the *enoyl-CoA hydratase and 3-hydroxyacyl CoA dehydrogenase* (*EHHADH*) gene, exhibits greater specificity for long-chain dicarboxylic acids [78,79].

The final cleavage step of peroxisomal β-oxidation produces acetyl-CoA (for non-BCFAs) or propionyl-CoA (for BCFAs), along with a shortened acyl-CoA chain. This reaction is catalyzed by either peroxisomal 3-ketoacyl-CoA thiolase (ACAA1) or sterol carrier protein x (SCPx), a splice variant of sterol carrier protein 2 (SCP2). SCPx plays a particularly important role in the catabolism of BCFAs, such as pristanic acid, DHCA, and THCA [80,81]. However, the substrate specificity of ACAA1 remains unclear.

The shortened acyl-CoAs undergo successive rounds of β-oxidation in peroxisomes until medium-chain acyl-CoAs are generated. At this stage, both acetyl-CoA and medium-chain acyl-CoAs are exported from peroxisomes and transferred to mitochondria for complete oxidation into carbon dioxide and water. This transfer can occur via two pathways: (i) as free metabolites, or (ii) in the form of acetyl- or acylcarnitines. In the first pathway, acyl-CoA thioesterases (ACOTs) hydrolyze acetyl-CoA and medium-chain acyl-CoA thioesters to produce free acetate and medium-chain fatty acids, which can then diffuse freely into mitochondria for further oxidation [82]. In the second pathway, carnitine acetyltransferase (CRAT) and carnitine octanoyltransferase (CROT) convert acetyl-CoA and medium-chain acyl-CoAs into acetylcarnitine and acylcarnitine, respectively. CRAT also mediates the conversion of propionyl-CoA into propionylcarnitine [83]. These acylcarnitines then cross the peroxisomal membrane and enter mitochondria via the carnitine shuttle system for further metabolism.

The oxidation of unsaturated FAs in peroxisomes involves three auxiliary enzymes: (i) peroxisomal 2,4-dienoyl-CoA reductase 2 (DECR2) (Figure 4B), (ii) 2-enoyl-CoA isomerase (ECI2), also known as peroxisomal Δ^3^,Δ^2^-enoyl-CoA isomerase (PECI) (Figure 4C), and (iii) Δ^3,5^,Δ^2,4^-enoyl-CoA isomerase (ECH1) (Figure 4D). The specific enzyme(s) recruited depend(s) on the position and number of the double bonds in the FA substrate. FAs with even-numbered double bonds require the concerted action of DECR2 and ECI2, whereas those with odd-numbered double bonds can be processed either by ECI2 alone or through a combined pathway involving all three enzymes [84,85]. DECR2 is exclusively localized to peroxisomes, while ECI2 and ECH1 are present in both peroxisomes and mitochondria [86,87,88]. Although both organelles can metabolize unsaturated FAs, they exhibit distinct substrate specificities. Peroxisomes preferentially oxidize unsaturated VLCFAs, such as erucic acid (C22:1 n-3) and arachidonic acid (C20:4 n-6), while mitochondria primarily oxidize unsaturated long-chain FAs [89,90]. Notably, tetracosahexaenoic acid (C24:6 n-3), in its CoA activated form, undergoes a single round of peroxisomal β-oxidation, yielding DHA (C22:6 n-3), which is subsequently exported from peroxisomes and incorporated in various biological pathways [91,92].

Peroxisomal β-oxidation also contributes to the chain-shortening of dicarboxylic acid CoA esters, such as azelaic acid (C9) and hexadecanedioic acid (C16) [78,93]. These dicarboxylic acids are produced via the ω-oxidation of FAs and are imported into peroxisomes in their CoA-activated forms by the ABCD3 transporter [94,95]. Once inside, β-oxidation shortens these substrates, generating intermediates such as succinyl-CoA (C4). These products can then be transferred to mitochondria for further metabolism, either as free acids or as carnitine conjugates, such as succinylcarnitine. Within peroxisomes, acyl-CoA thioesterase 4 (ACOT4), also known as peroxisomal succinyl-CoA thioesterase, plays a key role in hydrolyzing succinyl-CoA to free succinate. This succinate can subsequently enter mitochondria, where it is further metabolized through the tricarboxylic acid (TCA) cycle [83].

#### 3.3.2. α-Oxidation

The primary role of peroxisomal α-oxidation is the breakdown of phytanic acid (Figure 5), a 3-methyl BCFA that cannot undergo β-oxidation due to the presence of a methyl group at the β-carbon (position 3). Phytanic acid is primarily derived from dietary sources such as dairy products, red meat, and certain plants. During α-oxidation, the first carbon (position 1) is removed as formyl-CoA, shifting the methyl group to the α-carbon (position 2). This structural modification unblocks the β-carbon, enabling the resulting pristanic acid to undergo subsequent degradation via β-oxidation [96].

Phytanoyl-CoA can enter peroxisomes directly via the ABCD3 transporter. Alternatively, it can be generated within the peroxisome from phytenoyl-CoA through the action of peroxisomal trans-2-enoyl-CoA reductase (PECR), with phytenoyl-CoA also entering peroxisomes via ABCD3. Once inside, phytanoyl-CoA is hydroxylated by phytanoyl-CoA 2-hydroxylase (encoded by *PHYH*) to form 2-hydroxyphytanoyl-CoA. This reaction requires 2-oxoglutarate and produces succinate and CO_2_ as by-products. Subsequently, 2-hydroxyphytanoyl-CoA is cleaved by 2-hydroxyacyl-CoA lyase 1 (encoded by *HACL1*), yielding pristanal and formyl-CoA. Pristanal is then oxidized by an NAD^+^-dependent aldehyde dehydrogenase to form pristanic acid, which is activated by very long-chain acyl-CoA synthetase (ACSVL1), officially known as SLC27A2. Pristanoyl-CoA then undergoes three cycles of β-oxidation, forming either propionyl-CoA (if the 2-methyl branch is present) or acetyl-CoA (if absent), which are transported to mitochondria for further metabolism [85].

#### 3.3.3. Peroxisomal Fatty Acid Oxidation and Prostate Cancer

FA oxidation has emerged as a crucial metabolic process that supports PCa cell survival, progression, and resistance to therapy [97,98,99,100]. Early studies provided initial evidence for the involvement of (peroxisomal) FA oxidation in PCa pathogenesis by demonstrating the increased expression and activity of AMACR, also known as P504S, in PCa tissues (Figure 1B) [101,102,103,104,105]. The consistent differential expression of AMACR between malignant and benign prostate tissue makes it both a diagnostic and prognostic biomarker [106,107,108,109]. In addition, functional studies have further validated AMACR’s critical role in PCa progression [104,110,111,112,113]. Despite the well-established clinical relevance of AMACR in PCa, its dual localization to both peroxisomes and mitochondria presents a notable knowledge gap [75,76,77]. While both organelles participate in β-oxidation, they differ in substrate specificity and perform non-redundant functions. This highlights the need to delineate the specific contributions of each AMACR pool in the context of PCa.

It is also important to note that while AMACR has been identified as an androgen-independent growth modifier in PCa [104], the mechanisms linking elevated AMACR protein levels to altered PCa cell behavior remain unclear. One prevailing hypothesis posits that AMACR functions as a gatekeeper of β-oxidation, and that its overexpression enables PCa cells to shift their metabolic reliance toward FA β-oxidation, thereby supporting tumor progression [109,111,114]. Alternatively, given AMACR’s key role in the metabolism of BCFAs and bile acids [85], and the fact that some of these metabolites are high-affinity ligands for multiple orphan nuclear receptors (e.g., androstane receptor, pregnane X receptor, farnesoid X receptor, and peroxisome proliferator-activated receptors) that are implicated in cancer [115,116], increased AMACR activity may also modulate PCa cell growth through nuclear receptor signaling pathways.

The central role of AMACR in PCa has prompted further investigation into additional enzymes involved in peroxisomal BCFA oxidation. Notably, the increased expression and activity of key players such as ACOX3 and HSD17B4, responsible for the initial and subsequent steps of 2-methyl BCFA β-oxidation, have been observed in PCa tissues [38,104]. In conjunction with AMACR upregulation, these findings underscore the potential importance of BCFA metabolism in PCa. Moreover, single nucleotide polymorphisms in HSD17B4 have been linked to higher Gleason scores and poor clinical outcomes [117,118,119,120]. Functional studies have further demonstrated the critical role of HSD17B4 in driving tumorigenic phenotypes in PCa cell lines such as LNCaP, PC3, and DU145. Interestingly, the post-translational regulation of HSD17B4 stability through acetylation at lysine 669—modulated by sirtuin 3 (SIRT3) and CREB-binding protein (CREBBP) [121]—provides a potential mechanism for fine-tuning HSD17B4 activity in the context of cancer.

Interestingly, Sharifi and colleagues reported that HSD17B4 isoform 2 possesses additional enzymatic activity, catalyzing the oxidation of testosterone and dihydrotestosterone at their 17β-OH position, thereby converting them into their inactive 17-keto-forms [122]. The overexpression of this isoform suppressed tumor growth in LAPC4 xenografts in castrated mice, whereas its knockdown enhanced AR signaling and promoted tumor growth in 22Rv1 xenografts. Notably, these effects were observed exclusively under castration conditions, aligning with clinical observations that isoform 2 is downregulated in CRPC [122].

Building on the established role of FA β-oxidation in PCa metabolism, Mills and colleagues identified ECI2 as a key enzyme involved in the β-oxidation of unsaturated FAs, which is notably overexpressed in PCa tissues [98]. The knockdown of ECI2 reduced cell proliferation and induced cell death in PCa cells. Metabolomic analyses further revealed a decline in lactate production, decreased levels of Krebs cycle intermediates, and lipid accumulation [98]. Importantly, ECI2 Δ^3^,Δ^2^-enoyl-CoA isomerase activity is present in both mitochondria and peroxisomes [88].

In a related study, Butler and colleagues investigated DECR2, a peroxisomal enzyme involved in the β-oxidation of unsaturated FAs [97]. DECR2 expression was found to be elevated in metastatic CRPC, and its silencing induced cell cycle arrest, reduced tumor growth in vivo, and altered cellular lipid profiles. These findings, alongside previous research, emphasize the critical role of unsaturated FA β-oxidation in driving PCa malignancy [97].

Arachidonic acid and its metabolites have also been implicated in PCa development by promoting tumor growth, inflammation, and metastasis [123,124,125]. Peroxisomes are capable of oxidizing unsaturated VLCFAs, including arachidonic acid [90]. However, their specific role in regulating arachidonic acid levels within the context of PCa remains unclear. In addition, the synthesis of DHA, which requires coordinated activity between enzymes in the ER and peroxisomal β-oxidation [92], has been consistently associated with the inhibition of PCa progression [126,127,128,129,130]. Despite this, the contribution of peroxisomal β-oxidation to DHA synthesis in PCa has yet to be fully elucidated.

The disruption of peroxisomal β-oxidation, achieved by knocking out acyl-CoA oxidases, significantly reduces the metastatic capacity of PC3-mm2 cells in vivo, without affecting primary tumor growth. This disruption also results in intracellular triglyceride accumulation in vitro [131]. Moreover, peroxisomal VLCFA β-oxidation is elevated in PCa cell lines (22Rv1 and PC3) compared to non-malignant PNT1A cells [39]. Silencing CRAT reduced tumorigenic properties under low-serum conditions, and the downregulation of ACOX or CRAT was associated with a decreased activation of calcium/calmodulin-dependent kinase II (CaMKII), a kinase frequently upregulated in metastatic PCa [131]. Additional studies have further emphasized the role of peroxisomal FA oxidation in PCa pathophysiology by revealing altered expression patterns of peroxisome-associated proteins and key enzymes involved in FA oxidation [35,38,39,97].

### 3.4. Peroxisomes, Redox Homeostasis, and Prostate Cancer

#### 3.4.1. Peroxisomes and Redox Homeostasis

Peroxisomes play a central role in maintaining redox homeostasis [12]. The term “peroxisomes” was coined by Christian de Duve to reflect the organelle’s dual role in both generating and degrading H_2_O_2_ [132]. This naming was inspired by the co-sedimentation of CAT with H_2_O_2_-producing enzymes during subcellular fractionation.

Peroxisomes contain a variety of flavin-dependent oxidases that generate H_2_O_2_ as a metabolic by-product [12]. While H_2_O_2_ was traditionally viewed as merely a harmful oxidative species, accumulating evidence supports its function as a crucial signaling molecule [133]. Through sulfenome analysis and redox electrophoretic mobility shift assays, our lab has identified over 400 cellular targets of peroxisome-derived H_2_O_2_ [134]. These targets span various functional categories, including antioxidant enzymes, kinases, phosphatases, calcium-binding proteins, metabolic enzymes, and transcription factors [134,135].

CAT, the most abundant enzyme in peroxisomes, plays a central role in decomposing H_2_O_2_ due to its exceptionally high catalytic rate (Vmax = 587,000 μmol H_2_O_2_/μmol heme/s), despite its relatively low substrate affinity (Km = 80 mM) [136]. CAT is targeted to peroxisomes via a non-canonical -KANL motif, a weak peroxisomal targeting signal. However, this weak motif results in low-affinity binding to the cytosolic receptor PEX5, whose redox-sensitive nature can further impair import under oxidative stress. As a result, a significant portion of CAT remains in the cytosol, particularly during conditions of elevated oxidative burden [137].

In addition to CAT, peroxisomes harbor glutathione S-transferase kappa 1 (GSTK1), the only known peroxisomal enzyme that utilizes glutathione and also exhibits glutaredoxin activity [138]. Peroxisomes further contain epoxide hydrolase 2 (EPHX2), which catalyzes the conversion of FA-derived epoxides into less reactive dihydrodiols [139]. Another notable peroxisomal enzyme is mitochondrial amidoxime-reducing component 2 (MARC2, encoded by *MTARC2*), a molybdoenzyme capable of reducing a variety of N-oxygenated compounds. Recent evidence also indicates that MARC2 can reduce H_2_O_2_ in HEK-293T cells [140,141], suggesting a broader role in redox regulation.

Peroxisomes also contain superoxide dismutase 1 (SOD1), which catalyzes the dismutation of superoxide anions into H_2_O_2_ [142]. In addition, two peroxiredoxins (PRDXs), PRDX1 and PRDX5, are associated with peroxisomes. PRDX1 primarily reduces H_2_O_2_ [143], whereas PRDX5, an atypical peroxiredoxin, acts on a broader spectrum of substrates, including alkyl hydroperoxides, peroxynitrite, and, to a lesser extent, H_2_O_2_ [144]. Notably, PRDX1 has a much lower Km for H_2_O_2_ compared to CAT, making it especially effective at mitigating oxidative stress under low H_2_O_2_ conditions [136].

It is important to note that peroxisomes do not harbor unique antioxidant enzymes. Instead, the antioxidant machinery found in peroxisomes is also shared with other cellular compartments. For example, CAT and EPHX2 are also present in the cytosol [137,139], while GSTK1 is localized to peroxisomes, mitochondria, and the ER [138]. MARC2 is found in both peroxisomes and mitochondria [145,146], and SOD1, PRDX1, and PRDX5 are distributed across peroxisomes, mitochondria, the nucleus, and the cytosol [142,144,147].

Depending on the cell type, physiological state, and environmental context, peroxisomes can act as either sources or sinks of H_2_O_2_.

#### 3.4.2. Peroxisomes as Redox Regulators in Prostate Cancer

Although peroxisomes play a vital role in maintaining redox homeostasis, their specific function in PCa, a disease characterized by elevated ROS levels, remains incompletely understood [148]. CAT, the most abundant peroxisomal enzyme, has been shown to be significantly downregulated in PCa tissues compared to adjacent normal prostate tissue, as evidenced by early immunohistochemical analyses [149,150,151], a pattern that aligns with CAT mRNA expression data from TCGA (Figure 1C). This reduction in CAT expression correlates with elevated intracellular H_2_O_2_ levels [152], suggesting a disruption in peroxisomal antioxidant capacity. Furthermore, meta-analyses have associated the *CAT C262T* gene polymorphism, which reduces CAT enzymatic activity, with an increased risk of PCa, highlighting a potential genetic basis for redox imbalance in the disease [153,154,155].

AR activation downregulates CAT expression in PCa cells through the Forkhead Box O3A (FOXO3A)-ROS axis [156] and similarly reduces both CAT expression and activity in LNCaP cells [35]. Several studies have investigated the effects of altering CAT expression in PCa cell lines. One study demonstrated that CAT knockout suppressed the proliferation of PC3 cells in vitro and reduced xenograft growth [157]. In contrast, another study reported that siRNA-mediated CAT silencing increased proliferation in 22Rv1 and LNCaP cells [35]. The latter findings are consistent with earlier reports in breast cancer and mouse aortic endothelial cells, where reduced CAT expression was linked to enhanced cell proliferation [158,159]. Similarly, CAT overexpression has been shown to inhibit cell proliferation in both smooth muscle and MCF-7 breast cancer models [160,161].

The pharmacological inhibition of CAT in PCa has also been studied [35,157,162,163,164]. BT-1, a dual inhibitor of SOD1 and CAT, increased intracellular ROS and superoxide anion levels, resulting in cell cycle arrest, apoptosis, and reduced tumor growth in DU145 xenografts, while the irreversible CAT inhibitor 3-amino-1,2,4-triazole (3-AT) elevated ROS levels without affecting superoxide radicals [162]. In silico analyses revealed that BT-1, a benzaldehyde thiosemicarbazone derivative, binds to a site on CAT distinct from the heme active site targeted by 3-AT [162]. At high concentrations (250 mM), 3-AT inhibits PC3 cell proliferation [157], while lower concentrations (10 mM) reduce CAT activity without impacting proliferation [35]. In addition, pretreatment with 3-AT sensitizes Rv1 and PC3 cells to apoptosis induced by benzyl isothiocyanate [163]. More recently, BT-Br, another benzaldehyde thiosemicarbazone derivative, has been developed to target the NADP-binding site of CAT, differing from 3-AT’s heme-active site binding [164]. BT-Br treatment elevates ROS levels, induces autophagy and cell death, and suppresses tumor growth in DU145 xenografts [164]. Although these inhibitors show promising therapeutic potential, their specificity and possible off-target effects require further investigation.

PRDX1, a primary target of peroxisome-derived H_2_O_2_ [134], supports tumor growth and angiogenesis in PC3-M xenografts by promoting vascular endothelial growth factor (VEGF) expression in a Toll-like receptor 4 (TLR4)-dependent manner [165]. The knockdown of PRDX1 leads to reduced hypoxia-inducible factor-1α (HIF-1α) levels and mitogen-activated protein kinase (MAPK) 1/2 [166]. PRDX1 also directly interacts with AR [167,168], and its silencing disrupts AR transactivation in LNCaP cells [169]. Furthermore, PRDX1 cooperates with thioredoxin domain-containing 9 (TXNDC9) to regulate AR signaling [170]. Notably, the downregulation of PRDX1 enhances AR inhibition and suppresses cell proliferation [171]. PRDX1 activity is further amplified through its interaction with tumor protein D52, and silencing either or both proteins impairs PCa cell growth [172]. In addition, Holliday junction recognition protein (HJURP) has been shown to enhance PRDX1 activity, thereby reducing lipid peroxidation and protecting against ferroptosis [173].

PRDX5 influences resistance to AR inhibitors and may play a role in the progression to castration resistance. The inhibition of PRDX5 suppresses the proliferation of drug-tolerant persister cells in vitro, delays CRPC progression in animal models, and helps stabilize prostate-specific antigen (PSA) levels as well as metastatic lesions in patients [174]. Meanwhile, SOD1 is crucial for the C-X-C motif chemokine ligand 12 (CXCL12)-mediated activation of Akt signaling and contributes to resistance against etoposide-induced apoptosis [175]. EPHX2, an enzyme involved in the metabolism of epoxyeicosatrienoic acids—bioactive lipid mediators derived from arachidonic acid—is typically downregulated in PCa but shows higher expression in metastatic tumors compared to localized cases [176]. Silencing EPHX2 in LNCaP cells, but not in VCaP cells, reduces cell viability, AR signaling, and PSA expression [177]. Furthermore, EPHX2 is part of a six-gene signature that predicts disease-free survival in PCa [178].

When examining peroxisomal β-oxidation and H_2_O_2_ production (Figure 3), it is important to recognize that this pathway targets acyl-CoAs rather than free FAs. Consequently, variations in substrate availability or β-oxidation activity can affect the cellular redox balance by altering the levels of free CoASH, a crucial intracellular non-protein thiol [179,180]. Notably, the mRNA expression of genes involved in peroxisomal H_2_O_2_ production—such as ACOX2, pipecolic acid oxidase (PIPOX), and hydroxyacid oxidase 1 (HAO1)—as well as those regulating the CoASH/acyl-CoA ratio, including ACOT8 and SLC25A17 (also known as PMP34, a peroxisomal transporter of nicotinamide adenine dinucleotide (NAD^+^), flavin adenine dinucleotide (FAD), and free CoASH [181]), is altered in prostate adenocarcinoma compared to normal prostate tissue (Figure 1C). Although the functional consequences of these changes are not yet fully understood, recent studies have linked ACOX2 downregulation and HAO1 upregulation with biochemical recurrence in PCa [182]. Reduced circulating levels of PIPOX in patients with metastatic PCa have been linked to impaired sarcosine metabolism and a decreased oncogenic potential [183]. In contrast, ACOT8—a peroxisomal acyl-CoA thioesterase that hydrolyzes fatty acyl-CoAs to release CoASH and free FAs—is upregulated and has been implicated in tumorigenesis through the inhibition of ferroptosis [184]. Furthermore, transcriptome analyses have identified SLC25A17 as one of the most highly upregulated genes in enzalutamide-resistant PCa cells, where it contributes to the development of antiandrogen resistance [185].

## 4. Future Directions

In this review, we have highlighted the alterations in peroxisomes observed in PCa and examined their potential roles in disease initiation and progression. Studying peroxisome function in PCa presents several significant challenges. One major obstacle is the multi-compartmental localization of many peroxisome-associated proteins, which are often found in several cellular compartments, including the mitochondria, ER, nucleus, and cytosol. For example, AMACR, a well-established player in PCa, is localized to both peroxisomes and mitochondria [75,76,77]. While its involvement in BCFA metabolism is well recognized, the specific contributions of its peroxisomal versus mitochondrial forms to cancer progression remain unclear. Similar complexity exists with enzymes such as ECI2 and CAT, which are dually localized to peroxisomes and either mitochondria or the cytosol, respectively [88,137].

Adding to this complexity, several metabolic pathways implicated in PCa, such as ether lipid and PUFA biosynthesis, depend on functional interactions between peroxisomes and other organelles, including the ER [11,92]. These overlapping and compartmentalized roles make it challenging to determine which organelle is primarily responsible for specific oncogenic alterations. Therefore, a deeper understanding of the peroxisome–organelle crosstalk is essential to elucidate their mechanistic contributions to PCa progression and to identify novel therapeutic targets.

Importantly, alterations in peroxisome function within prostate cells are likely to play a key role in the metabolic reprogramming that drives tumor initiation and progression. Exploiting the functional differences between normal and malignant prostate cells could thus represent a promising strategy for therapeutic development. However, despite mounting evidence of peroxisomal dysregulation in primary prostate tumors and experimental models, many studies remain largely correlative and do not establish a clear causal relationship. In particular, it remains unclear whether peroxisomal dysfunction actively promotes tumorigenesis or is a consequence of other oncogenic events.

Changes in mRNA levels do not always correspond consistently to the protein levels of peroxisomal enzymes in PCa. Furthermore, enzymatic activity is frequently not assessed, and when it is, the assays often fail to differentiate between closely related isoforms, such as ACOX2 and ACOX3. While large-scale public databases like Gene Expression Omnibus (GEO) Profiles offer valuable information, they often lack independent experimental validation, partly due to the heterogeneity of PCa, which encompasses multiple subtypes, forms, and stages. These challenges underscore the urgent need for a systematic and comprehensive characterization of peroxisomal alterations in patient-derived samples and across diverse PCa models.

While in vitro models provide valuable insights, they often fail to fully capture the metabolic complexity of the tumor environment. Factors such as oxygen availability, nutrient gradients, androgen signaling, and the absence of stromal and immune components can significantly influence peroxisome-related processes [186,187,188]. These limitations raise concerns about the physiological relevance of in vitro findings and underscore the need for more contextually accurate models to study peroxisomal function in PCa. In this context, although peroxisomal metabolism is recognized as a key contributor to immunometabolism [189], its impact on immune cell infiltration in PCa remains largely unclear. However, emerging evidence points to a potential connection between peroxisomal metabolism and the tumor immune microenvironment. For example, bioinformatics analyses of RNA-seq data from TCGA revealed an inverse correlation between AMACR expression and the infiltration of CD4^+^ T cells, macrophages, and neutrophils [190]. Similarly, a computational study identified a six-gene lipid metabolism signature—including SLC27A2 (ACSVL1)—that is associated with immune cell infiltration in localized PCa [191]. Moreover, elevated ACOX1 expression in exhausted CD8^+^ T cells has been linked to IL-8 secretion by PCa-derived exosomes, which activates PPARγ in these T cells [192]. Furthermore, the potential link between peroxisomal alterations and PCa stemness remains entirely unexplored to date. Understanding whether peroxisome-associated metabolic pathways contribute to the maintenance or behavior of PCa stem cells represents an important area for future research.

Methodological limitations remain a significant challenge in the study of peroxisome function. Chemical inhibitors are sometimes used to investigate enzyme activity, yet concerns about their specificity persist. For example, the widely used CAT inhibitor 3-AT not only targets CAT but also interferes with the α-oxidation of 3-methyl-BCFAs like phytanic acid [193], heme biosynthesis [194], and cytochrome P450 2E1 (CYP2E1) activity [195]. Discrepancies between chemical inhibition and genetic silencing, such as the divergent effects of CAT inhibition and knockdown on LNCaP cell proliferation [35], highlight the need for more selective and well-validated experimental tools. Similarly, the frequent use of non-specific ROS probes like 2′,7′-dichlorodihydrofluorescein diacetate (H_2_DCFDA) can lead to misleading interpretations due to their limited specificity for H_2_O_2_ [196]. In contrast, genetically encoded biosensors such as roGFP2-Orp1 and Hyper7 offer greater specificity and enable the spatially resolved monitoring of H_2_O_2_ dynamics within distinct subcellular compartments, including peroxisomes [197].

Emerging evidence indicates that disruptions in peroxisomal metabolism may significantly contribute to PCa biology. However, it remains uncertain whether these alterations actively drive tumorigenesis or represent secondary adaptations to other oncogenic processes, underscoring the need for further investigation. Future research should aim to pinpoint the specific peroxisomal enzymes, metabolic pathways, and interorganelle interactions that are dysregulated in PCa. This includes validating findings across diverse experimental models and patient-derived samples, improving the specificity of functional assays, and leveraging advanced imaging and biosensing technologies—particularly to examine the influence of AR signaling on peroxisome homeostasis. Ultimately, gaining a detailed mechanistic understanding of peroxisome biology in PCa could inspire new avenues for the development of targeted therapeutic therapies.

## 5. Conclusions

Peroxisomal alterations are increasingly recognized as key contributors to the metabolic landscape of PCa. Disruptions in peroxisomal function—particularly in fatty acid oxidation, ether lipid synthesis, and redox regulation—appear to drive the metabolic reprogramming characteristic of the disease. These changes are closely associated with tumor progression and the emergence of therapeutic resistance. Deepening our understanding of these peroxisome-related pathways will be crucial for elucidating the molecular underpinnings of PCa biology and may ultimately pave the way for more effective therapeutic interventions.

## Figures and Tables

**Figure 1 cancers-17-02243-f001:**
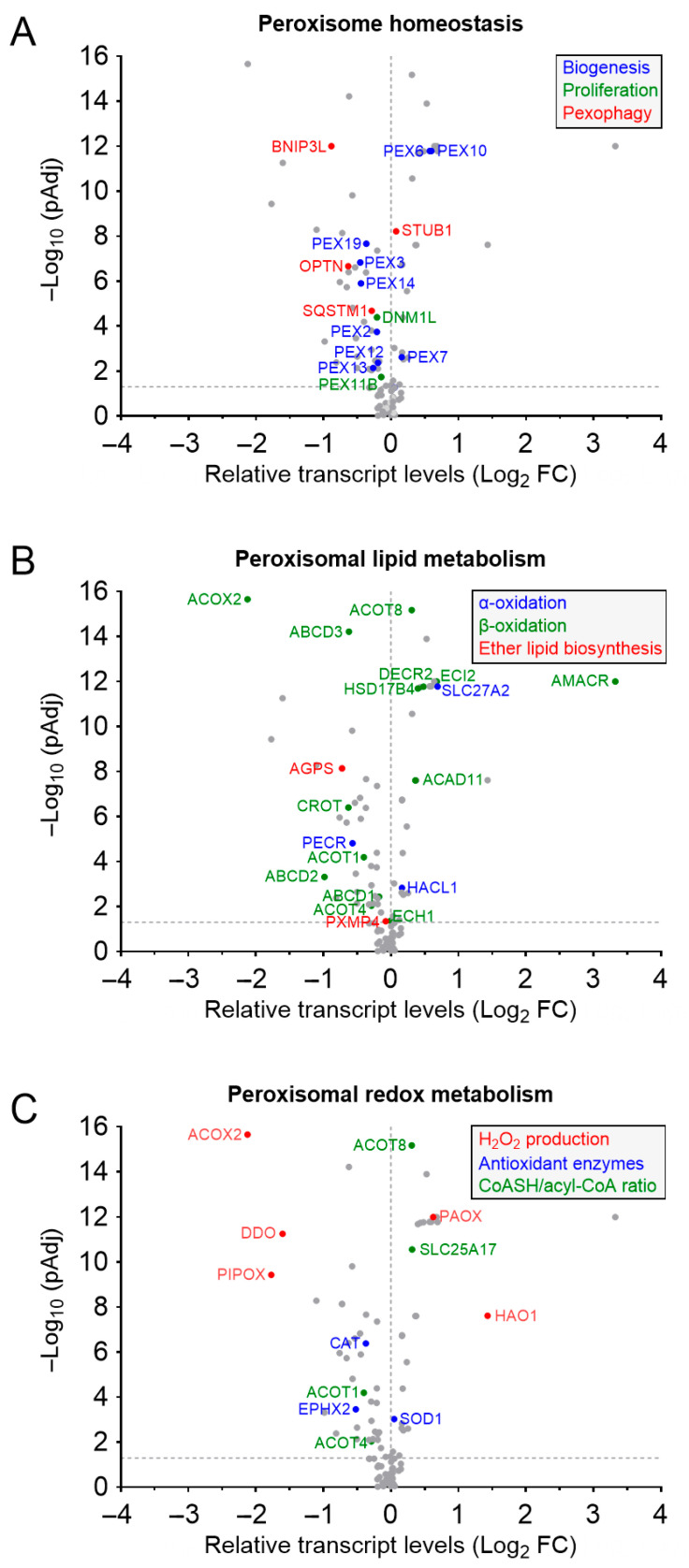
Differential expression of peroxisome-related genes in primary prostate adenocarcinoma compared to normal tissue. Volcano plots display differential gene expression between primary prostate adenocarcinoma (*n* = 497) and normal prostate tissue (*n* = 52): genes encoding proteins involved in (**A**) peroxisome homeostasis, (**B**) peroxisomal lipid metabolism, and (**C**) peroxisomal redox metabolism. Only genes with statistically significant changes in expression (adjusted *p*-value < 0.05; −Log_10_ adjusted *p*-value > 1.3; indicated by a dotted horizontal line) are labeled. Data were obtained from The Cancer Genome Atlas Program (TCGA) mRNA expression profiles of primary prostate cancer and normal tissue, accessed via the University of Alabama at Birmingham Cancer Data Analysis portal (UALCAN; https://ualcan.path.uab.edu/; accessed on 12 March 2025).

**Figure 2 cancers-17-02243-f002:**
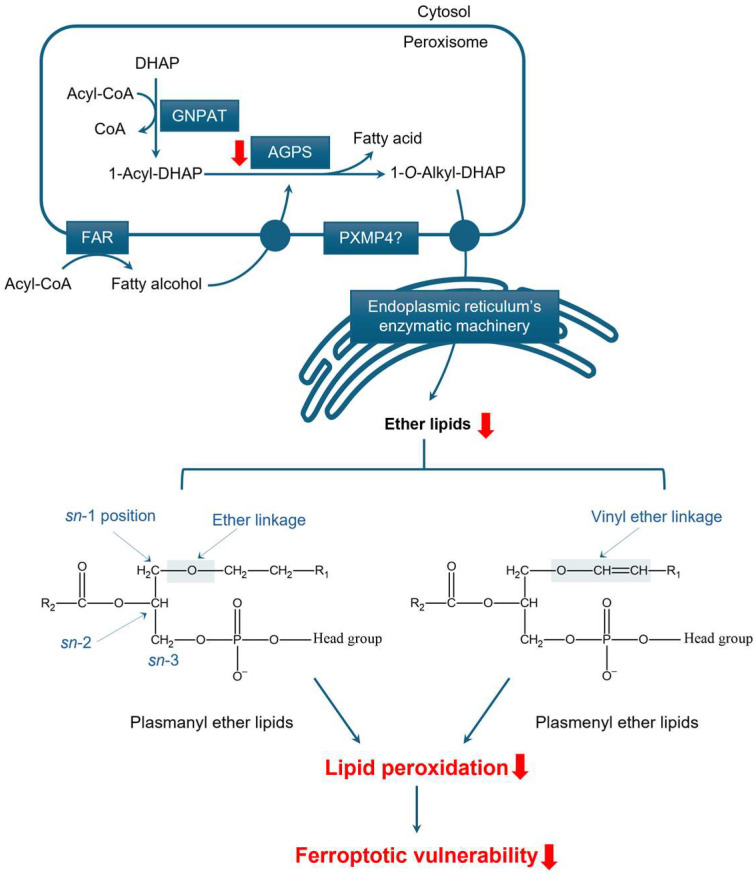
Peroxisomal involvement in ether lipid synthesis. The pathway initiates in peroxisomes, where glycerone-3-phosphate *O*-acyltransferase (GNPAT) catalyzes the formation of 1-acyl-dihydroxyacetone-3-phosphate (1-acyl-DHAP) from DHAP and acyl-CoA. 1-alkyl-glycerone-3-phosphate synthase (AGPS) then replaces the acyl moiety with a fatty alcohol, produced by fatty acyl-CoA reductase (FAR), to form the ether bond in 1-*O*-alkyl-DHAP. This intermediate is subsequently transferred to the ER for further processing into ether lipids. Although peroxisomal membrane protein 4 (PXMP4) is predicted to act upstream of, or within, the ether lipid metabolic process, its exact function remains unclear. Structural differences at the *sn*-1 position between plasmanyl and plasmenyl ether lipids are highlighted. Downregulation in PCa is represented by a red arrow pointing downward.

**Figure 3 cancers-17-02243-f003:**
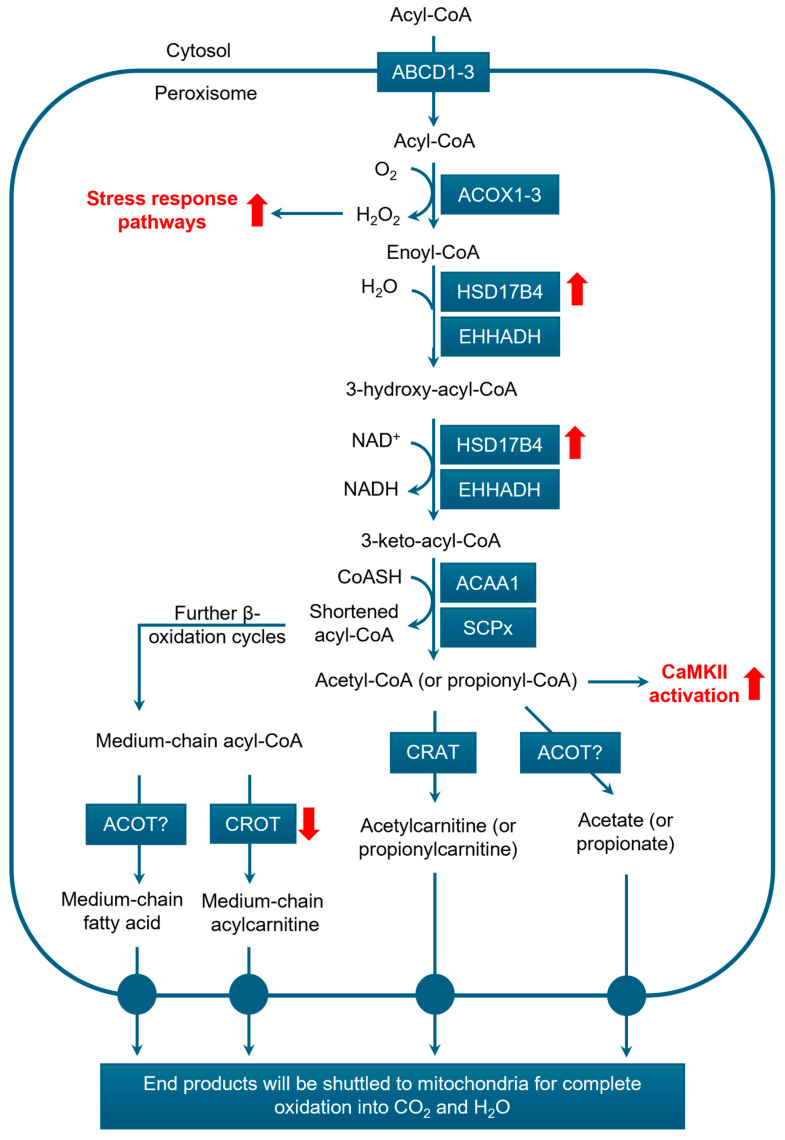
Schematic representation of the peroxisomal β-oxidation machinery. The diagram depicts the key transporters, enzymes, and metabolic steps involved in the peroxisomal β-oxidation pathway. Upregulation and downregulation in PCa are represented by red upward and downward arrows, respectively. Notable components include ATP binding cassette subfamily D (ABCD), peroxisomal 3-ketoacyl-CoA thiolase (ACAA1), acyl-CoA oxidase (ACOX), acyl-CoA thioesterase (ACOT), calcium/calmodulin-dependent kinase II (CaMKII), carnitine acetyltransferase (CRAT), carnitine octanoyltransferase (CROT), hydroxysteroid 17-beta dehydrogenase 4 (HSD17B4), enoyl-CoA hydratase and 3-hydroxyacyl CoA dehydrogenase (EHHADH), and sterol carrier protein x (SCPx).

**Figure 4 cancers-17-02243-f004:**
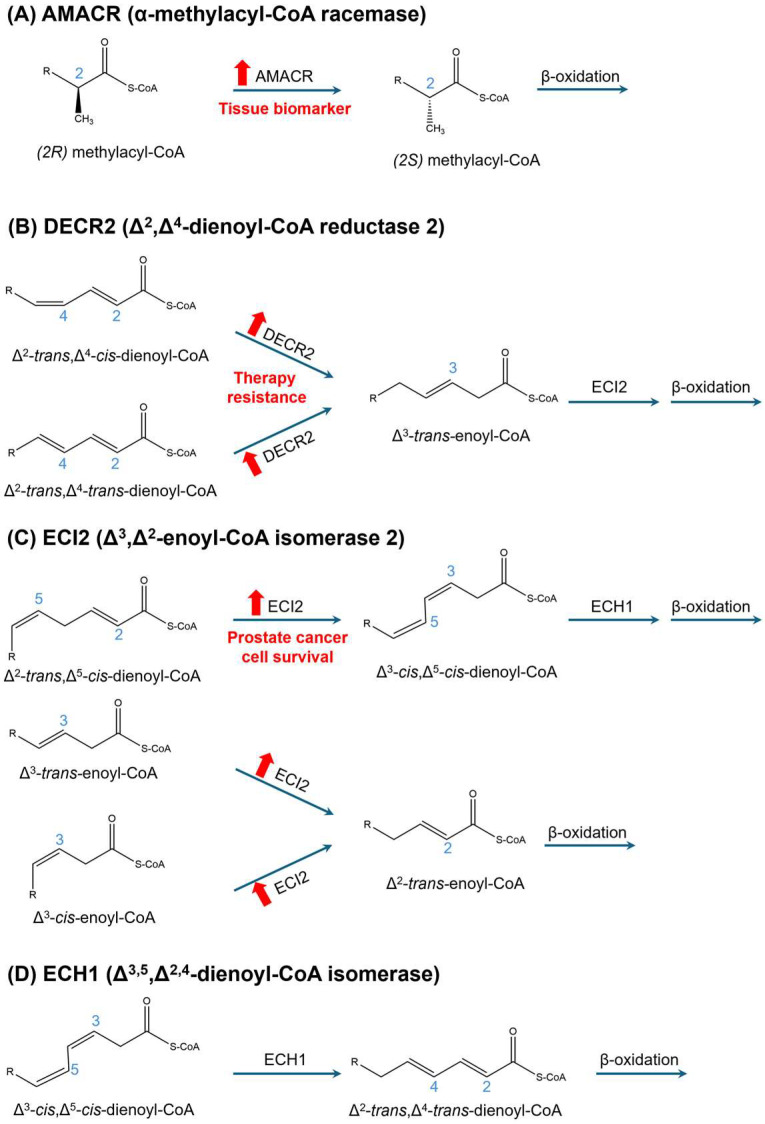
Auxiliary enzymes involved in peroxisomal β-oxidation. This schematic illustrates the chemical reactions catalyzed by (**A**) α-methylacyl coenzyme A racemase (AMACR), (**B**) 2,4-dienoyl CoA reductase 2 (DECR2), (**C**) 2-enoyl-CoA isomerase (ECI2), and (**D**) Δ^3,5^,Δ^2,4^-enoyl-CoA isomerase (ECH1) in the peroxisomal β-oxidation pathway. AMACR is essential for the β-oxidation of 2-methyl-branched-chain fatty acids, whereas DECR2, ECI2, and ECH1 participate in the β-oxidation of unsaturated fatty acids. Red upward arrows represent upregulation.

**Figure 5 cancers-17-02243-f005:**
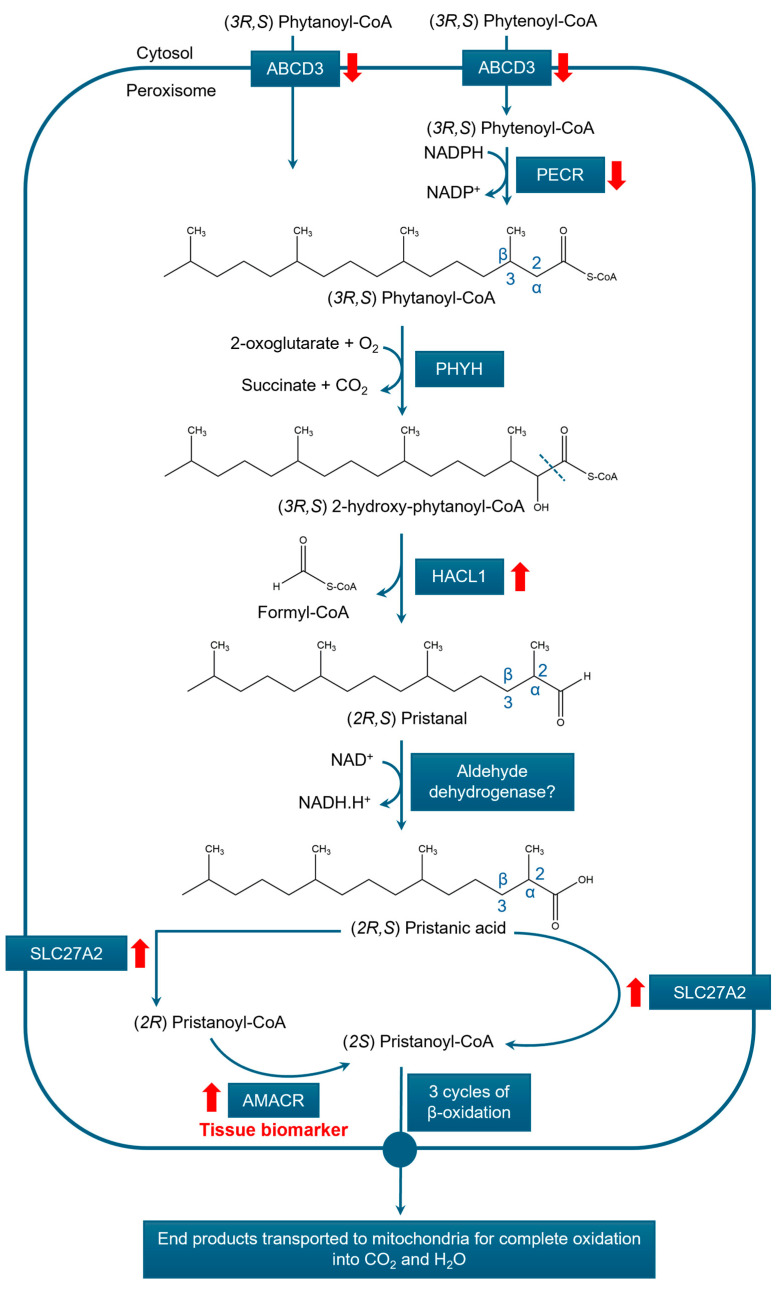
Schematic overview of the enzymatic steps involved in the peroxisomal α-oxidation of phytanic acid. Key enzymes include ATP-binding cassette subfamily D member 3 (ABCD3), α-methylacyl coenzyme A racemase (AMACR), 2-hydroxyacyl-CoA lyase 1 (HACL1), peroxisomal trans-2-enoyl-CoA reductase (PECR), and phytanoyl-CoA 2-hydroxylase (PHYH). Upregulation and downregulation in PCa are indicated by red upward and downward arrows, respectively.

## Data Availability

The mRNA expression data (displayed in Figure 1) for primary prostate cancer and normal tissues were retrieved from The Cancer Genome Atlas (TCGA) via the University of Alabama at Birmingham’s Cancer Data Analysis Portal (UALCAN; https://ualcan.path.uab.edu/; accessed on 12 March 2025) [26].

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
