# Peer review of "Peroxisomal Alterations in Prostate Cancer: Metabolic Shifts and Clinical Relevance"

_cancers, 2025, doi:10.3390/cancers17132243_

Round 1
Reviewer 1 Report
Comments and Suggestions for Authors
The manuscript commences with an examination of the essential metabolic functions of peroxisomes, encompassing fatty acid oxidation, ether lipid synthesis, and redox regulation. This is followed by a discourse on their pertinence to prostate cancer (PCa). However, the current structure dedicates extensive portions of the early sections to introducing basic peroxisomal biology, which detracts from the central theme of the manuscript—namely, the significance of peroxisomal dysfunction in PCa. Consequently, the presentation's duration seems excessive.
Specifically, the manuscript provides redundant explanations of the fundamental functions of enzymes and transporters, such as ACOX, AMACR, and ACBD5, when they are first introduced. These functions are then discussed again in the context of cancer in the manuscript's latter half. This repetitive structure detracts from rather than enhances the reader's understanding.
Consequently, a comprehensive reorganization is strongly advised:
- The manuscript should be restructured to focus on peroxisome-associated proteins and metabolic pathways relevant to prostate cancer. Examples of relevant proteins include ACOX family, AGPS, CAT, PRDX1/5, and SLC25A17. The manuscript should explain the functional roles and pathological implications of these proteins in that order. In other words, the sequence should follow "cancer relevance → background function" to better capture and retain the reader's interest.
- Long descriptions of genetic disorders such as X-linked adrenoleukodystrophy and Refsum disease should be minimized or removed, as they are not directly related to PCa. Similarly, detailed explanations of α-oxidation and ABC transporters should be condensed or integrated only if their relevance to PCa is clearly established.
- The majority of current figures primarily illustrate general fatty acid metabolism or schematic representations of transporters. However, they do not provide a visual explanation of how these pathways are connected to prostate cancer pathophysiology. These should be replaced with integrative pathway diagrams that explicitly depict the causal relationships between key peroxisomal players and cancer-associated phenotypes (e.g., proliferation, metastasis, therapeutic resistance). This applies to Figures 1, 3, 4, 5, and 6.
- Figure 2 fails to visually convey its relevance to cancer based on its title or axis labels, and therefore fails to function effectively as a figure for cancer research. The significance thresholds, such as Log₂ FC > ±1 and −log(p-adj) > 1.3 (FDR < 0.05), are not clearly marked, which makes interpretation difficult. Although changes in AMACR and AGPS are included in the plot, their biological importance in tumorigenesis is not readily apparent from the figure itself. Additionally, the references to Figure 2 in the main text are vague and insufficient.
Author Response
The figure, line, and reference numbers correspond to those in the revised manuscript.
Reviewer 1
The manuscript commences with an examination of the essential metabolic functions of peroxisomes, encompassing fatty acid oxidation, ether lipid synthesis, and redox regulation. This is followed by a discourse on their pertinence to prostate cancer (PCa). However, the current structure dedicates extensive portions of the early sections to introducing basic peroxisomal biology, which detracts from the central theme of the manuscript—namely, the significance of peroxisomal dysfunction in PCa. Consequently, the presentation's duration seems excessive.
We appreciate the reviewer’s feedback regarding the manuscript structure and the emphasis on general peroxisomal biology in the early sections. In response, we have extensively revised the manuscript to provide a more focused and concise presentation of the background, with greater emphasis on the relevance of peroxisomal functions in the context of prostate cancer (PCa). The introduction to basic peroxisomal biology has been shortened and streamlined, and the transition to disease relevance now occurs earlier and more directly. These structural changes aim to more closely align the narrative with the manuscript’s central theme: the role of peroxisomal dysfunction in PCa. Details of the revisions are provided in the specific points outlined below.
Specifically, the manuscript provides redundant explanations of the fundamental functions of enzymes and transporters, such as ACOX, AMACR, and ACBD5, when they are first introduced. These functions are then discussed again in the context of cancer in the manuscript's latter half. This repetitive structure detracts from rather than enhances the reader's understanding.
We have revised the manuscript to remove redundancy. The functions of enzymes and transporters such as ACOX, AMACR, ACBD5, and others are now explained only at their first mention. For example, the initial description of ACOX appears on lines 375-380, AMACR on lines 380-383; and ACBD5 on lines 363-366.
The manuscript should be restructured to focus on peroxisome-associated proteins and metabolic pathways relevant to prostate cancer. Examples of relevant proteins include ACOX family, AGPS, CAT, PRDX1/5, and SLC25A17. The manuscript should explain the functional roles and pathological implications of these proteins in that order. In other words, the sequence should follow "cancer relevance → background function" to better capture and retain the reader's interest.
We appreciate the reviewer’s suggestion to restructure the manuscript by prioritizing peroxisome-associated proteins and pathways according to their cancer relevance. In response, we have substantially reduced the section on peroxisome biogenesis and degradation and removed the corresponding Figure. In addition, we have reorganized other sections to highlight the cancer relevance of peroxisomal pathways earlier in the manuscript. However, we respectfully disagree with the proposed sequence of presenting cancer relevance prior to introducing the fundamental biological context. We believe that a clear understanding of processes such as plasmalogen biosynthesis and peroxisomal β- and α-oxidation is essential to fully appreciate their implications in PCa. Presenting pathological relevance without first establishing the functional context may confuse readers unfamiliar with peroxisomal metabolism. Therefore, we have retained a logical progression where the relevant background information is provided first, followed by its direct connection to PCa. We hope the reviewer appreciates and agrees with this rationale.
Long descriptions of genetic disorders such as X-linked adrenoleukodystrophy and Refsum disease should be minimized or removed, as they are not directly related to PCa. Similarly, detailed explanations of α-oxidation and ABC transporters should be condensed or integrated only if their relevance to PCa is clearly established.
The revised document now includes only a single sentence summarizing genetic disorders associated with defects in peroxisomal lipid metabolism (see lines 348-350). We believe this provides the minimum essential information needed to inform readers outside the peroxisome field that peroxisomal lipid metabolism is critical for overall organismal health, and that genetic – and, as our review further demonstrates, other – disturbances in this process can contribute to disease. As illustrated in the revised Figure 5, disturbances in α-oxidation and ABCD3 function are indeed linked to PCa and are relevant to its pathology. We trust that our reorganization of the manuscript, particularly in Section 3.3.3., more clearly highlights these important connections.
The majority of current figures primarily illustrate general fatty acid metabolism or schematic representations of transporters. However, they do not provide a visual explanation of how these pathways are connected to prostate cancer pathophysiology. These should be replaced with integrative pathway diagrams that explicitly depict the causal relationships between key peroxisomal players and cancer-associated phenotypes (e.g., proliferation, metastasis, therapeutic resistance). This applies to Figures 1, 3, 4, 5, and 6.
We have modified the figures as requested to enhance clarity and focus. However, as mentioned above, the original Figure 1 was removed to streamline the review. In addition, as highlighted in the Future Directions section (lines 705-709), although there is increasing evidence of peroxisomal dysregulation in primary prostate tumors and experimental models, many studies remain largely correlative and do not establish definitive causal relationships. Specifically, it is still unclear whether peroxisomal dysfunction actively promotes tumorigenesis or is a consequence of other oncogenic events.
Figure 2 fails to visually convey its relevance to cancer based on its title or axis labels, and therefore fails to function effectively as a figure for cancer research. The significance thresholds, such as Log₂ FC > ±1 and −log(p-adj) > 1.3 (FDR < 0.05), are not clearly marked, which makes interpretation difficult. Although changes in AMACR and AGPS are included in the plot, their biological importance in tumorigenesis is not readily apparent from the figure itself. Additionally, the references to Figure 2 in the main text are vague and insufficient.
Regarding Figure 2 (corresponding to Figure 1 in the revised manuscript), we would like to clarify the following:
- The UALCAN database does not provide explicit fold-change (FC) cutoff values, as its primary aim is to facilitate interactive exploration of gene expression data rather than to perform standardized differential expression analysis based on strict thresholds. As such, FC cutoffs are not displayed, as their interpretations in isolation may be misleading without a broader biological context. We now explicitly note this in the manuscript (see lines 129-135)
- Concerning the FDR < 0.05 threshold, all panels include a dotted horizontal line representing this cutoff. To improve clarity, we now also explicitly mention this in the figure legend (see lines 193-195).
- Regarding the figure citations in the main text, the figure is first referenced in line 148 (panel A), with subsequent references at lines 272, 277, and 481 (panel B), and lines 610 and 673 (panel C). As these panels form a cohesive figure, we chose to present the full figure at its first citation (panel A), while referring to individual panels later in the text where relevant data are discussed in detail.
We believe these clarifications, together with the revisions made to the legend and citations, enhance the interpretability and relevance of the figure within the context of PCa research.
Reviewer 2 Report
Comments and Suggestions for Authors
The review article entitled “cancers-3680679_ Peroxisomal alterations in prostate cancer: metabolic shifts and clinical relevance”, submitted to the section “Molecular Cancer Biology” of the special issue “Advancements in Molecular Research of Prostate Cancer”, addresses the underexplored role of peroxisomes in prostate cancer (PCa), emphasizing their involvement in tumor metabolism, progression, and therapy resistance. The article provides a timely and relevant synthesis of emerging evidence that places peroxisomes alongside mitochondria and the endoplasmic reticulum as key organelles in cancer biology. The discussion focuses particularly on the unique metabolic features of PCa, which diverge from the typical Warburg phenotype and rely instead on lipid metabolism and oxidative phosphorylation.
While the review presents a clear and informative overview of peroxisomal functions and their alterations in PCa, several methodological and structural aspects require improvement to enhance its academic rigor and utility for future research.
First, the timeframe covered by the review is not specified, which limits its integration with past and future reviews on the topic. Additionally, the criteria for article selection, the number of articles reviewed, and the databases consulted are not reported. These are essential elements of transparency and reproducibility, especially in a critical review aiming to inform current and future scientific work. The omission of these elements may compromise the review’s clarity and reduce its potential for indexing and citation within bibliometric platforms.
Although the introduction includes relevant background literature and effectively contextualizes the role of peroxisomes in cellular metabolism, it would benefit from the explicit statement of a working hypothesis prior to the presentation of the review’s aims. Doing so would provide a clearer conceptual framework and align the review more closely with hypothesis-driven scholarship.
Moreover, the absence of a dedicated methodology section is a critical limitation. In the current scientific landscape, where reviews serve as essential tools for synthesizing rapidly evolving knowledge, it is imperative to clearly describe the methodological approach. This includes inclusion/exclusion criteria, search strategies, and data extraction methods. Such information is not only important for evaluating the reliability of the conclusions but also for enabling replication and comparison across reviews.
The results of the review are well-structured by thematic sections, which aids comprehension given the biochemical complexity of the topic. The figures are generally effective in summarizing key mechanisms; however, Figure 2 lacks sufficient resolution, impairing readability. If this figure is reproduced from a previously published source, appropriate permissions must be verified and clearly indicated, in accordance with editorial standards.
While the article does not include a formal discussion section, it does provide a “Future Directions” section. Nevertheless, a more detailed assessment of the methodological quality of the studies reviewed would strengthen the critical nature of the work. The article notes that “methodological limitations remain a significant challenge in the study of peroxisome function,” yet does not systematically address how these limitations affect the conclusions drawn.
The conclusion effectively summarizes the review’s main findings and responds to the objectives stated in the introduction. However, the manuscript would benefit from greater methodological transparency and a clearer critical framework in order to maximize its impact.
In summary, the review is engaging and addresses a novel and underexplored topic with potential implications for cancer therapy. However, to meet the standards of a high-impact academic review, clear methodological detailing and a more rigorous critical appraisal of the literature are required.
Author Response
The figure, line, and reference numbers correspond to those in the revised manuscript.
Reviewer 2
We are grateful for the reviewer’s thoughtful insights and trust that the responses below adequately address the concerns raised.
First, the timeframe covered by the review is not specified, which limits its integration with past and future reviews on the topic. Additionally, the criteria for article selection, the number of articles reviewed, and the databases consulted are not reported. These are essential elements of transparency and reproducibility, especially in a critical review aiming to inform current and future scientific work. The omission of these elements may compromise the review’s clarity and reduce its potential for indexing and citation within bibliometric platforms.
According to the journal’s author guidelines, narrative reviews are expected to include an abstract, introduction, body sections covering relevant content, a conclusion and future directions section, and a references section. While a detailed methodology is not typically required for narrative reviews, in deference to the reviewer’s constructive comments, we have added a dedicated Methodology section to enhance transparency and reproducibility (see lines 101-135). This section outlines the data sources and search strategy, the eligibility and exclusion criteria, the study screening and selection process, and the data extraction procedure. We hope this addition addresses the reviewer’s concerns and strengthens the clarity and bibliometric robustness of the review.
Although the introduction includes relevant background literature and effectively contextualizes the role of peroxisomes in cellular metabolism, it would benefit from the explicit statement of a working hypothesis prior to the presentation of the review’s aims. Doing so would provide a clearer conceptual framework and align the review more closely with hypothesis-driven scholarship.
In response to the reviewer’s request, we have revised the end of the Introduction to include a clear conceptual hypothesis and framework. Specifically, we now state: “This comprehensive literature review is guided by the working hypothesis that alterations in peroxisome function are not merely correlative but play a mechanistic role in PCa pathogenesis and treatment outcomes. We critically examine how peroxisomal dysfunction influences tumor biology with particular emphasis on its interaction with AR signaling.” (see lines 96-100). We believe this addition strengthens the overall focus of the review and provides a solid conceptual foundation for the discussions that follow.
Moreover, the absence of a dedicated methodology section is a critical limitation. In the current scientific landscape, where reviews serve as essential tools for synthesizing rapidly evolving knowledge, it is imperative to clearly describe the methodological approach. This includes inclusion/exclusion criteria, search strategies, and data extraction methods. Such information is not only important for evaluating the reliability of the conclusions but also for enabling replication and comparison across reviews.
See above.
The results of the review are well-structured by thematic sections, which aids comprehension given the biochemical complexity of the topic. The figures are generally effective in summarizing key mechanisms; however, Figure 2 lacks sufficient resolution, impairing readability. If this figure is reproduced from a previously published source, appropriate permissions must be verified and clearly indicated, in accordance with editorial standards.
Regarding Figure 1 in the revised manuscript (previously Figure 2), we clarify that the data were obtained from The Cancer Genome Atlas (TCGA) via the University of Alabama at Birmingham Cancer Data Analysis Portal (UALCAN), and were not reproduced from a previously published figure. In accordance with TCGA guidelines (https://shorturl.at/nnOq1), we now explicitly state in the Acknowledgements section “The mRNA expression data shown in this study are in whole based on data generated by the TCGA Research Network (https://www.cancer.gov/tcga) via the University of Alabama at Birmingham Cancer Data Analysis Portal (UALCAN). In addition, we have updated the figure to improve its resolution and overall visual quality.
While the article does not include a formal discussion section, it does provide a “Future Directions” section. Nevertheless, a more detailed assessment of the methodological quality of the studies reviewed would strengthen the critical nature of the work. The article notes that “methodological limitations remain a significant challenge in the study of peroxisome function,” yet does not systematically address how these limitations affect the conclusions drawn.
We thank the reviewer for this important comment. According to the journal’s guidelines, narrative reviews do not include a formal Discussion section; instead, they feature a Future Directions section where key limitations and challenges in the current research landscape are addressed. In our revised manuscript, we specifically highlight major challenges in studying peroxisomal alterations in PCa (see lines 685-762). These limitations were carefully considered throughout our interpretation of the literature and are explicitly noted at relevant points in the text to guide the reader. For example, while discussing the significance of AMACR, we emphasize its dual localization to peroxisomes and mitochondria and stress the need for further studies to clarify which pool plays a more critical role in prostate cancer progression (see lines 484-488). Similarly, regarding the use of 3-amino-1,2,4-triazole (3-AT), a commonly used catalase inhibitor, we address concerns about its specificity and potential off-target effects, which may complicate the interpretation of results in studies examining peroxisomal function (see lines 741-746). We believe these targeted discussions, combined with the Future Directions section, provide a nuanced and critical assessment of methodological challenges and their impact on the conclusions drawn.
In summary, the review is engaging and addresses a novel and underexplored topic with potential implications for cancer therapy. However, to meet the standards of a high-impact academic review, clear methodological detailing and a more rigorous critical appraisal of the literature are required.
To address concerns about methodological transparency, we have added a dedicated Methodology section in the revised manuscript (see lines 101-135). This section outlines the general approach used to identify and analyze relevant literature, thereby enhancing the rigor and clarity of our review.
Reviewer 3 Report
Comments and Suggestions for Authors
The manuscript by Hussein et al. illustrates that a deeper understanding of peroxisome biology in prostate cancer may facilitate the development of novel therapies aimed at enhancing patient outcomes.
I have a few comments for polishing the manuscript.
- The authors should also state how mitochondrial respiration is also involved with cancer progression. The authors should review the following article in this regard
https://doi.org/10.1016/j.isci.2025.112219
- Can peroxisomal metabolic shifts affect the immune infiltrations in prostate cancer?
- The authors should state the major limitations associated with the study.
- Did peroxisomal alterations affect the stemness of PCa cells?
Author Response
The figure, line, and reference numbers correspond to those in the revised manuscript.
Reviewer 3
We sincerely thank the reviewer for the valuable suggestions to improve the manuscript and trust that our responses below effectively address the points raised.
The authors should also state how mitochondrial respiration is also involved with cancer progression. The authors should review the following article in this regard: https://doi.org/10.1016/j.isci.2025.112219
We thank the reviewer for emphasizing the importance of mitochondrial respiration in cancer progression. For this reason, we included specific examples in the Introduction Section illustrating how subcellular organelles contribute to cancer development. In particular, we highlight that alterations in mitochondrial outer membrane permeabilization and mitochondrial permeability transition enable malignant cells to evade regulated cell death pathways, thereby promoting tumor progression (see lines 54-58). This example underscores the critical role of mitochondrial integrity in sustaining cancer cell survival and metabolic adaptability.
Can peroxisomal metabolic shifts affect the immune infiltrations in prostate cancer?
We thank the reviewer for raising this important point. In response, we have added the following section to the Future Directions: "In this context, although peroxisomal metabolism is recognized as a key contributor to immunometabolism [189], its impact on immune cell infiltration in PCa remains largely unclear. However, emerging evidence points to a potential connection between peroxisomal metabolism and the tumor immune microenvironment. For example, bioinformatics analyses of RNA-seq data from TCGA revealed an inverse correlation between AMACR expression and infiltration of CD4⁺ T cells, macrophages, and neutrophils [190]. Similarly, a computational study identified a six-gene lipid metabolism signature – including SLC27A2 (ACSVL1) – that is associated with immune cell infiltration in localized PCa [191]. Moreover, elevated ACOX1 expression in exhausted CD8⁺ T cells has been linked to IL-8 secretion by PCa–derived exosomes, which activates PPARγ in these T cells [192].” (see lines 724-734).
The authors should state the major limitations associated with the study.
We appreciate the opportunity to clarify this aspect of our review. The major limitations of the current body of research are addressed in the Future Directions section of the manuscript. Specifically, we outline several key challenges in studying peroxisomal alterations in PCa, including (i) the compartment-specific localization of peroxisome-associated proteins (lines 686-694), (ii) the complexity of peroxisome-organelle interactions (lines 695-701), (iii) the predominantly correlative nature of existing studies, which limits mechanistic insight (lines 702-709), and (iv) methodological issues such as the specificity of chemical inhibitors and the technical challenges associated with ROS detection (lines 739-751). In addition, we highlight the limitations of conventional in vitro models and emphasize the need for more physiologically relevant system, along with validation using patient-derived samples (lines 719-724). Collectively, these points underscore the current gaps in the field and the need for more refined experimental approaches.
Did peroxisomal alterations affect the stemness of PCa cells?
In response to this question, we conducted a focused literature search to investigate a potential link between PCa stemness and peroxisomal alterations. However, to the best of our knowledge, no studies have directly examined this specific relationship to date. While both cancer stemness and peroxisomal metabolism have been independently studied in the context of PCa, their potential interplay remains largely unexplored. To acknowledge this important gap, we have added the following to the revised manuscript: "Furthermore, the potential link between peroxisomal alterations and PCa stemness remains entirely unexplored to date. Understanding whether peroxisome-associated metabolic pathways contribute to the maintenance or behavior of PCa stem cells represents an important area for future research” (see lines 734-738).
Round 2
Reviewer 1 Report
Comments and Suggestions for Authors I am in agreement with the acceptance of this paper as the authors have adequately addressed the concerns of my peer review.Reviewer 2 Report
Comments and Suggestions for Authors
I have carefully reviewed the revised version of the manuscript titled “cancers-3680679_ Peroxisomal alterations in prostate cancer: metabolic shifts and clinical relevance”, as well as the authors’ detailed response to the suggestions made in the previous review.
The authors have addressed each point thoroughly, providing clear clarification and improvements that significantly enhance the quality and comprehensibility of the review. I would like to commend them for their diligent work and the thoughtful revisions implemented.
Reviewer 3 Report
Comments and Suggestions for Authors
The manuscript by Hussein et al. illustrates that a deeper understanding of peroxisome biology in prostate cancer may facilitate the development of novel therapies aimed at enhancing patient outcomes.
The authors have addressed all the previous comments. Thus, the manuscript can be accepted in its present form.